# Analysis of Exon Skipping Applicability for Dysferlinopathies

**DOI:** 10.3390/cells14030177

**Published:** 2025-01-24

**Authors:** Jamie Leckie, Sebastian Hernandez Rodriguez, Martin Krahn, Toshifumi Yokota

**Affiliations:** 1Department of Medical Genetics, Faculty of Medicine and Dentistry, University of Alberta, Edmonton, AB T6G 2H7, Canada; jleckie@ualberta.ca (J.L.); antonios@ualberta.ca (S.H.R.); 2INSERM, Marseille Medical Genetics, U1251, Aix-Marseille University, APHM, 13385 Marseille, France; martin.krahn@ap-hm.fr; 3Medical Genetics Department, Timone Hospital, APHM, 13005 Marseille, France; 4The Friends of Garrett Cumming Research & Muscular Dystrophy Canada HM Toupin Neurological Sciences Research, Edmonton, AB T6G 2H7, Canada

**Keywords:** dysferlinopathy, exon skipping, antisense oligonucleotide (ASO), applicability

## Abstract

Exon skipping, mediated through antisense oligonucleotides (ASOs), is a promising approach to exclude pathogenic variants from the *DYSF* gene and treat dysferlinopathies. Understanding the applicability of various exon skipping strategies in the total patient population, an analysis not previously performed, can help guide researchers in prioritizing therapies with the broadest potential impact. Using data from the UMD-DYSF database, we evaluated all reported pathogenic variants in dysferlinopathy patients for the applicability of single- or double-exon skipping approaches to exclude the pathogenic variants while maintaining the open reading frame. A total of 61 theoretically applicable exon skipping strategies were identified, with the potential to address 90.0% of the pathogenic variants reported—44.6% through single-exon skipping and 45.3% through double-exon skipping. The most broadly applicable targets include exons 28 and 29 (9.0%), exons 27 and 28 (6.7%), and exons 50 and 51 (5.4%). While numerous theoretically applicable strategies were identified, it remains unclear if the truncated proteins produced through each exon skipping strategy will have improved functionality to alleviate patient symptoms. Further preclinical studies and clinical trials will be essential to determine the effectiveness of these therapies, potentially expanding access to disease-modifying treatments for dysferlinopathy patients.

## 1. Introduction

Dysferlinopathies comprise a group of rare muscle disorders caused by pathogenic variants in the *DYSF* gene that result in a broad spectrum of clinical phenotypes and disease severities [1]. These pathogenic variants result in significant or complete loss of functional dysferlin proteins, which in turn causes skeletal muscle damage and necrosis [2]. One promising therapeutic approach to bypass these variants is exon skipping, where the exon containing the variant, and adjacent exons, if necessary to maintain the open reading frame, are selectively excluded [3]. This strategy can be effectively mediated using antisense oligonucleotides (ASOs), which have already received FDA approval for the treatment of other rare diseases, such as spinal muscular atrophy and Duchenne muscular dystrophy (DMD) [4,5].

While early preclinical trials have shown encouraging results for ASO-mediated exon skipping in dysferlinopathies, further research is necessary before this strategy can be translated to clinical applications [6,7]. Pathogenic variants associated with the disease have been identified in all exons of the *DYSF* gene, and, due to the variant-specific nature of exon skipping strategies, each exon skipping strategy would only be applicable to a subset of the dysferlinopathy patient population. Thus, evaluating the applicability of all potential exon skipping strategies across the patient population is crucial to allow researchers to prioritize those that could benefit the largest number of individuals.

Currently, a comprehensive evaluation of potential exon skipping targets for dysferlinopathies has not been reported. This study aims to fill that gap by providing an in-depth analysis of the applicability of all possible single- and double-exon skipping approaches that could theoretically target patient pathogenic variants reported in the comprehensive UMD_DYSF database (http://www.umd.be/DYSF/. Accessed on 17 November 2024) [8]. Understanding the applicability of these strategies will help optimize therapeutic development and potentially expand treatment options for patients affected by dysferlinopathies. 

## 2. Background on Dysferlinopathies

Dysferlinopathies are rare autosomal recessive diseases caused by pathogenic variants in the *DYSF* gene that result in a significant reduction or complete absence of the dysferlin protein [9,10]. Dysferlin is a 237 kDa protein that plays a diverse range of roles and is primarily expressed in striated muscle [11,12,13]. Dysferlin is part of the ferlin protein family, along with myoferlin, otoferlin, Fer1L4, Fer1L5, and Fer1L6. Ferlin proteins are characterized by their large size and multiple tandem C2 domains, which typically form a binding pocket with an affinity for Ca^2+^ and phospholipids [14,15]. Dysferlin, specifically, is composed of seven C2 domains and is observed to localize predominantly in the sarcolemma or t-tubule membrane in uninjured muscle cells [16,17]. This protein is largely known for its role in muscle membrane repair [18]. Microlesions in the plasma membrane can occur because of mechanical stress associated with activity. Upon damage, dysferlin travels to the cytoplasm where, in a Ca^2+^-dependent manner, it regulates the trafficking and fusion of vesicles to create a patch for the disrupted membrane [16,19,20]. Dysferlin has also been observed to play an important role in t-tubule maintenance [16,21], regulating Ca^2+^ levels following mechanical stress [22,23], and has a variety of other functions [24,25,26]. The role of dysferlin in cardiac tissues is less understood at this time, but preliminary studies, and the similarities between cardiac myocytes and skeletal muscle cells, suggest that dysferlin plays a similar role in membrane repair and Ca^2+^ homeostasis in cardiac tissues [11,27,28,29]. 

Currently, the exact mechanisms underlying the pathogenesis of dysferlinopathies remain unclear. However, upon injury, dysferlin-deficient tissues have a diminished capacity to repair their membranes, which is believed to result in necrosis and the initiation of an inflammatory response that can damage surrounding tissue [26,30,31]. Muscle biopsies collected from patients with pathogenic *DYSF* variants typically present with a large variation in fiber size, featuring necrotic and regenerating fibers, alongside increased levels of connective and fatty tissue deposits and inflammatory cells [32,33,34,35]. Patients also present with elevated levels of muscle damage biomarkers, even before the onset of symptoms [36]. 

There is a diverse spectrum of dysferlinopathies that result from deficient or absent dysferlin proteins due to pathogenic variants in the *DYSF* gene. The two primary clinical phenotypes are Miyoshi muscular dystrophy (MMD) and limb-girdle muscular dystrophy R2 (LGMDR2), formerly known as LGMD2B. MMD is initially characterized by muscle weakness and atrophy in the distal muscles, particularly in the calves and plantar muscles, which later extends to the thighs and buttocks [37]. In contrast, LGMDR2 patients present with proximal muscle weakness, predominantly affecting the thighs and shoulders [38]. While the age of symptom onset varies significantly among patients, both MMD and LGMDR2 typically manifest around the age of 20 and progress slowly [1,39]. As the diseases advance, patients experience difficulty with walking, stair climbing, and often become reliant on canes or wheelchairs [40]. Less commonly, *DYSF* variants can also result in other dysferlinopathies such as distal myopathy with anterior tibial onset (DMAT), the proximodistal phenotype, and asymptomatic hyperCKemia. DMAT is associated with the fastest disease progression, beginning with weakness in the anterior tibial muscles and later spreading to the lower and upper limbs [36,37]. Patients with the proximodistal phenotype present with symptoms characteristic of both MMD and LGMDR2, with weakness affecting both distal and proximal muscles simultaneously [41]. A hallmark of all dysferlinopathies is a pronounced elevation in serum creatine kinase (CK) levels [42,43]. In cases of asymptomatic hyperCKemia, patients exhibit elevated CK levels without apparent muscle weakness, although there remains a potential risk for muscle involvement over time [37,44]. Despite attempts to uncover clear genotype–phenotype correlations, predicting disease phenotypes remains challenging due to the variability seen across different patient populations [45,46,47]. Significant phenotypic differences are observed not only among individuals with the same pathogenic *DYSF* variants but even among family members, suggesting that additional genetic or environmental factors may influence the symptoms and severity of disease experienced by individuals [47,48,49]. This complexity underscores the need for further research to better understand the underlying mechanisms driving these phenotypic differences. 

Currently, there are no curative therapies available for dysferlinopathies [2]. Patients primarily rely on symptomatic treatments, such as walking aids, physiotherapy, occupational therapy, and ankle or foot orthoses to improve mobility and overall quality of life [50]. However, numerous studies have focused on evaluating a range of different molecular and genetic therapies aiming to improve the patient’s phenotype and prevent progression [51]. These approaches include strategies like gene therapy, delivering a modified dysferlin gene using AAV vectors [52,53,54], and gene editing, using CRISPR-Cas9 systems to correct specific *DYSF* variants [55,56]. However, the AAV vectors used to deliver these therapies have significant limitations and safety concerns. Natural exposure to AAVs has resulted in 30-60% of the population developing antibodies against these vectors, which can compromise their effectiveness and raise the risk of adverse immune reactions [57,58,59]. Additional safety risks include the development of innate and adaptive immune responses and the potential genomic consequences of viral genome insertion [60,61]. Moreover, despite the therapeutic potential of AAV vectors, several AAV-related patient fatalities, often associated with high doses, have raised concerns about the clinical safety of these vectors [62,63]. Alternatively, ASO-mediated exon skipping, which uses non-viral approaches to exclude pathogenic variants from the final transcript, shows significant promise and will be discussed in further detail.

## 3. ASO-Mediated Exon Skipping

ASOs represent a promising therapeutic approach that can be designed to be capable of selectively modulating gene expression at the RNA level [4]. They are short (~20 base pairs in length), synthetic, single-stranded molecules that can bind through Watson–Crick base pairing to a complementary RNA sequence. Upon hybridization, ASOs can either alter the transcript’s splicing or prevent its expression [64,65]. However, delivering unmodified ASOs to target tissues is challenging due to their high susceptibility to degradation by nucleases and lysosomes, as well as their poor cellular uptake [66,67,68]. To address these challenges, chemical modifications have been introduced to enhance both the stability and cellular uptake of ASOs [67,68]. Some ASOs are modified to include a phosphorothioate (PS) backbone and/or the addition of a functional group at the 2′ position of the ribose sugar, which can help protect the ASO from nuclease degradation [69]. Additionally, alternative backbone chemistries, such as the development of phosphorodiamidate morpholino oligomers (PMOs), replacing the natural backbone with morpholino rings linked by phosphorodiamidate bonds, offer advantages like increased targeting affinity and improved safety [5,69]. By selectively incorporating these chemical modifications, ASOs can be tailored to function as versatile therapeutic tools with different mechanisms of action [70]. These mechanisms can be classified into three main categories: (1) RNAse H-dependent mRNA degradation; (2) splicing modulation via steric hindrance of key regulatory sites, such as splice acceptor/donor sites, exonic splicing enhancers (ESEs), or exonic splice silencers (ESSs), thereby blocking spliceosome machinery; and (3) translation modulation by obstructing regulatory elements, like translation inhibitory elements in 5′ UTRs [71,72,73]. 

ASO-mediated exon skipping has emerged as a promising therapeutic strategy for treating various genetic neurodegenerative and muscular dystrophy disorders by allowing for the exclusion of pathogenic variants from the final mRNA transcript [74,75]. The primary goal of this approach is to generate a truncated, yet partially functional, protein capable of restoring some degree of protein function [76]. This is achieved by using ASOs, with all nucleotides chemically modified, to induce alterations in the transcript’s splicing mechanisms [77]. By targeting specific exons carrying or flanked by small pathogenic variants, ASOs can facilitate the exclusion of these regions while preserving the reading frame [78,79,80]. In addition, exon skipping can also address disruptions in the reading frame caused by large deletions or insertions by promoting the skipping of adjacent exons, which restores a continuous reading frame across the transcript [81]. 

The location and type of pathogenic variant can significantly affect the success of exon skipping therapies, as the compatibility of the remaining exons after skipping depends on their intron phase, categorized by their position between codons [82,83]. Therefore, ASOs for exon skipping need to be carefully designed to ensure that the exclusion of the target exon or exons preserves the correct reading frame to maintain the functional integrity of the resulting protein [82,83,84]. Symmetrical exons, flanked by introns of the same phase, can generally be skipped without disrupting the protein’s reading frame. In contrast, asymmetrical exons, surrounded by introns of differing phases, present a greater challenge. Skipping only the affected asymmetrical exon is insufficient to restore transcript functionality. In such cases, a multi-exon skipping strategy, where both the mutated exon and adjacent exons are skipped, may be required to maintain a continuous reading frame and ensure proper protein expression (Figure 1) [85].

Given the significant therapeutic potential of ASO-mediated exon skipping, several ASOs have already received FDA approval for rare genetic disorders, with additional clinical trials ongoing to explore the effectiveness of alternative ASO therapies [86]. Notably, several ASOs have been approved for the treatment of DMD, including Eteplirsen, Viltolarsen, Golodirsen, and Casimersen [87]. By promoting the exclusion of specific exons, these therapies have demonstrated effectiveness in restoring partially functional dystrophin protein levels in patients who otherwise lack functional dystrophin [86,88]. Despite their potential, the clinical effectiveness of these therapies for DMD remains controversial at this time. The clinical significance of the minimal functional benefits observed is difficult to determine due to small patient cohorts and reliance on historical controls [79,89,90]. Enhancing ASO delivery and cellular uptake will be essential to improving their therapeutic effect [67].

Furthermore, multi-exon skipping (targeting two or more exons) has recently gained increased attention, as it offers a broader therapeutic application for a wider range of pathogenic variants, including those involving small variants on asymmetrical exons [91,92,93]. Preclinical studies in DMD animal models have shown that combining ASOs targeting different exons (ASO cocktails) is highly effective in restoring dystrophin production [94]. Despite these promising results, further clinical studies are required to confirm the efficacy of this approach in patients. 

### ASO-Mediated Exon Skipping for the Treatment of Dysferlinopathies

Currently, there are no FDA-approved exon skipping therapies available to patients with dysferlinopathies. However, previous studies have identified patients with variants in the *DYSF* gene that naturally induce exon skipping without disrupting the reading frame, resulting in partially functional dysferlin and a milder disease phenotype [95,96]. This observation suggests that ASO-mediated exon skipping therapy could be a promising approach for treating dysferlinopathies [95,96]. By targeting the disease-causing exon, and additional adjacent exons if necessary, ASOs can preserve the *DYSF* reading frame while excluding pathogenic variants, potentially leading to a less severe phenotype [97]. Furthermore, in cases involving large multi-exon deletions in the *DYSF* gene [98], a strategy similar to current DMD therapies could be applied, where promoting the skipping of adjacent exons could restore the *DYSF* reading frame [99], potentially enabling the production of a partially functional dysferlin protein.

Preliminary in vitro studies of ASO-mediated exon skipping strategies have shown promising results that warrant further study. Specifically, 2′-O-methyl (2′-OME) PS ASOs targeting the splice donor site of exon 32 were able to promote exon 32 skipping in patient-derived cells harboring variants within this exon [7]. The induced exon skipping led to the production of low levels of a quasi-dysferlin protein, which partially restored membrane repair capacity [7]. In another study, a novel pathogenic variant within intron 50 caused aberrant pseudo-exon inclusion. In this case, 2′-OME PS ASOs targeting the exonic splicing enhancer within intron 50 effectively prevented abnormal splicing, thereby increasing wild-type transcript levels and enhancing dysferlin protein expression in treated cells [6]. Additionally, multi-exon skipping strategies have been evaluated for dysferlinopathies, particularly to bypass pathogenic variants in asymmetrical exons. PMO cocktails targeting exons 28 and 29 successfully induced the skipping of both exons, resulting in a truncated but functional dysferlin protein that restored membrane-resealing capabilities in patient cells [97]. These promising findings indicate that ASOs can affectively promote exon skipping to rescue dysferlin production and restore cellular function. However, further preclinical and clinical studies are essential to translate these advances into viable therapies for this currently underserved patient population. 

A critical challenge associated with the effectiveness of these approaches is the possibility that certain exons contain essential regions, and their exclusion may result in a non-functional or improperly folded protein unable to restore its function [87,93]. For instance, while skipping exon 25 maintains the reading frame, it still produces an unstable product that fails to yield a functional protein [94]. Additionally, plasma-membrane-resealing assays have identified exons 19 to 21, 20 to 21, and 46 to 48 as crucial regions for dysferlin-associated membrane repair [95]. Thus, excluding these regions is unlikely to improve symptoms in patients. When feasible, analyzing previously reported in-frame large deletions in patient populations related to the target regions for exclusion may help predict the functionality of the truncated protein generated through exon skipping.

## 4. The Theoretical Applicability of ASO-Mediated Exon Skipping for the Treatment of Dysferlinopathies

Currently, there are no curative therapies for dysferlinopathies, and existing treatments are primarily focused on managing disease symptoms [2]. This review aims to provide a comprehensive overview of potential exon skipping therapies targeting known pathogenic *DYSF* variants recorded in the UMD-DYSF database [8]. Given that these therapies are variant-specific, a deeper understanding of each strategy’s applicability to the broader patient population can help guide research efforts toward the most impactful and widely applicable approaches.

### 4.1. Pathogenic Variants Associated with Dysferlinopathies

Variant analyses reveal a diverse spectrum of disease-causing pathogenic variants throughout the *DYSF* gene that lead to the complete absence of functional dysferlin in patient muscle tissues [100]. Data from various patient cohorts indicate that most cases involve missense and frameshift variants; however, unlike many genetic disorders, there are no identified mutational hotspots, regions where pathogenic variants frequently occur [8]. Interestingly, seven distinct founder variants have been reported [8]. A study of Portuguese patients from the northern interior region revealed that all seven patients examined carried a missense variant linked to a common founder effect, resulting in exon 49 skipping [101]. Additionally, some studies suggest that pathogenic variants may cluster within certain regions of the DYSF protein in specific populations [102,103]. For example, research on Chinese patients with dysferlinopathies identified a concentration of pathogenic variants within the N-terminal region of the gene, particularly between the C2C and C2B domains [103].

The UMD-DYSF database analyzed in this study is composed of 1174 pathogenic variants, of which 962 are reported in patients affected with dysferlinopathies and 212 are reported in relatives of the affected patients [8]. Only pathogenic variants reported in probands were evaluated during analysis. Of the 962 reported pathogenic variants in patients, 345 (35.9%) were associated with LGMDR2, 342 (35.6%) with MMD, 176 (18.3%) with LGMDR2 or Miyoshi myopathy, 38 (3.9%) with proximodistal phenotype, 20 (2.1%) with isolated hyperCKaemia, 11 (1.1%) with distal myopathy with anterior tibial onset (DMAT), 8 (0.8%) with pseudometabolic disease, 6 (0.6%) with exercise intolerance, 2 (0.2%) with congenital muscular dystrophy, 2 (0.2%) with rigid spine syndrome, 2 (0.2%) with stiffness after exercise, 2 (0.2%) with trunk and lower limb girdle stiffness, and 8 (0.8%) with an unknown diagnosis. Point mutations make up 54%, insertions/deletions make up 29%, and intronic variants make up 17% of the reported patient pathogenic variants in the UMD-DYSF database (Table 1). Among the point mutations, 49% are nonsense variants, while 51% are missense variants.

### 4.2. Theoretical Applicability of Exon Skipping for Dysferlinopathies

All 962 pathogenic variants reported in affected patients were evaluated for the theoretical applicability of single- and double-exon skipping strategies. These approaches were evaluated for their ability either to exclude intra-exonic disease-causing variants without disrupting the reading frame or to restore reading frame disruptions caused by intronic splice site variants. Splice site variants located nearest the 5′ end of the exon were presumed to result in skipping of the downstream exon and variants nearest the 3′ end were presumed to result in skipping of the upstream exon. Figure 2 is a schematic representation of the 55 exons that make up dysferlin proteins, and their associated intron phase, which can be used as a tool to visualize how the exon’s reading frame will fit together following exon skipping. 

Overall, 61 single- and double-exon skipping strategies were identified to be applicable for the UMD-DYSF patient population (Table 2). Of the 61 strategies, 39 involve skipping a single exon to exclude the pathogenic variants, and 22 of the strategies involve skipping two exons to exclude the variants and maintain the open reading frame. Due to the high distribution of disease-causing variants reported across the *DYSF* gene, there are no strategies that were found to be applicable for more than 9.0% of the patient population. The highest ranked exon skipping strategies involved exon 28 and 29 (9.0%), exon 27 and 28 (6.7%), and exon 50 and 51 (5.4%). Specifically, for single-exon skipping strategies, the most applicable targets identified include exon 11 (2.9%), exon 24 (2.8%), exon 37 (2.6%), exon 25 (2.6%), and exon 19 (2.6%). 

Many of the pathogenic variants reported were determined to have multiple different applicable strategies that have the potential to exclude the pathogenic variant and maintain or restore the open reading frame. For example, the c.2077delC variant in exon 22, which leads to a premature termination stop codon, can be excluded by targeting exons 21 and 22, or exons 22 and 23. If either of these strategies is translated to the clinic, the other strategy would become less applicable to the total population as a theoretically equally effective therapy would already be accessible to these patients. 

Overall, exon skipping was found to be applicable for 90.0% of the reported pathogenic *DYSF* variants in affected patients, as reported in the UMD-DYSF database (Table 3). Specifically, single-exon skipping has the potential to restore functional dystrophin expression for 44.6% of dysferlinopathy patients. Targeting two exons for exclusion was determined to be applicable for a further 45.3% of patients. The remaining 10% of pathogenic variants were not amenable to single- or double-exon skipping. These included splice site variants leading to the exclusion of an in-frame exon and pathogenic variants located in the first or last exon which cannot be addressed by exon skipping. They also included missense variants in exons 15, 18, 31, 33, and 44, which could potentially be excluded through multi-exon skipping (involving more than two exons) to maintain the reading frame.

### 4.3. The Clinical Translation of ASO-Mediated Exon Skipping for the Treatment of Dysferlinopathies

Although numerous strategies were identified to be theoretically applicable for the dysferlinopathy patient population, as reported in the UMD-DYSF database, none of these therapies have yet to receive FDA approval. In total, 61 strategies (39 single- and 22 double-exon skipping) were identified to be applicable. Figure 3 illustrates the total theoretical applicability of single- and double-exon skipping for the treatment of dysferlinopathies, highlighting the top ten exon skipping targets. A total of 44.6% of the reported pathogenic variants were determined to be amenable through single-exon skipping. Furthermore, 90.0% of the patient population was determined to have a pathogenic variant with the potential to be excluded through either single- or double-exon skipping. These findings confirm the incredible potential of ASO-mediated exon skipping to treat most patients with pathogenic *DYSF* variants resulting in disease. 

Despite the identification of numerous promising exon skipping strategies for dysferlinopathies, most approaches remain theoretical, as only a few have been evaluated in preclinical studies. As previously discussed, skipping certain critical regions of dysferlin may fail to improve the phenotype. All proposed strategies require comprehensive evaluation in vitro and/or in animal models before clinical translation [104]. While patient-derived myotubes are the most relevant cell type for assessing therapeutic effectiveness due to the therapy’s direct relevance to muscle pathology [7], fibroblasts from dysferlinopathy patients have also demonstrated utility in evaluating membrane repair capacity following ASO treatment [97]. Given the variant-specific nature of ASOs, ideal animal models would harbor variants amenable to the strategy under study. For example, MMex38 mice, which carry an exon 38 variant, were used to evaluate the efficacy of U7 small nuclear RNA-mediated skipping of exons 37 and 38 [105].

Among the most promising strategies identified for dysferlinopathies are double-exon skipping approaches, which would require a combination, or “cocktail”, of ASOs. However, these multi-exon skipping approaches face additional hurdles for clinical translation, as each ASO, along with their combined use, may necessitate individual toxicology testing, significantly increasing both time and costs associated with their development [106,107]. As a result, reforms in the drug approval process could potentially streamline their path to clinical use [85]. 

Although several ASOs have received FDA approval for other genetic disorders, as previously discussed, their cellular uptake in patients remains suboptimal, often requiring high-dose frequent injections to achieve therapeutic effect [108]. Depending on the delivery route, ASOs must transverse multiple biological barriers to reach target tissues and influence RNA [109]. To enhance their clinical translation potential, current preclinical studies are investigating systems to optimize ASO delivery and cellular uptake. Lipid-based nanoparticles (LNPs) have shown promise in enhancing ASO stability in the extracellular environment and promoting cellular uptake [110,111]. Other carriers under investigation include extracellular vesicles [112], dendrimers [113,114], metallic nanoparticles [115], and cell-penetrating peptides (CPPs) [109]. Notably, CPPs, which can be conjugated to neutrally charged PMOs, have received particular interest due to their ability to significantly increase ASO uptake in target tissues, where ASOs previously struggled to reach [116]. Continued investigation into delivery systems that can improve ASO efficacy at a reduced dose is expected to improve their potential for clinical translation [117].

## 5. Conclusions

Given the current lack of disease-modifying therapies for dysferlinopathies, ASO-mediated exon skipping has shown potential in bypassing disease-causing pathogenic variants to generate a truncated dysferlin protein with enhanced functionality [97]. This review identified numerous single- and double-exon skipping targets that could theoretically benefit 90% of patients with dysferlinopathy. However, further research is necessary to confirm whether the truncated proteins produced through these strategies are indeed functional before progressing to clinical trials [96]. Additionally, advancements in ASO chemical modifications, the development of delivery systems to enhance cellular uptake, and reforms to the drug approval process could significantly improve the potential for these therapies to become accessible to patients. 

## Figures and Tables

**Figure 1 cells-14-00177-f001:**
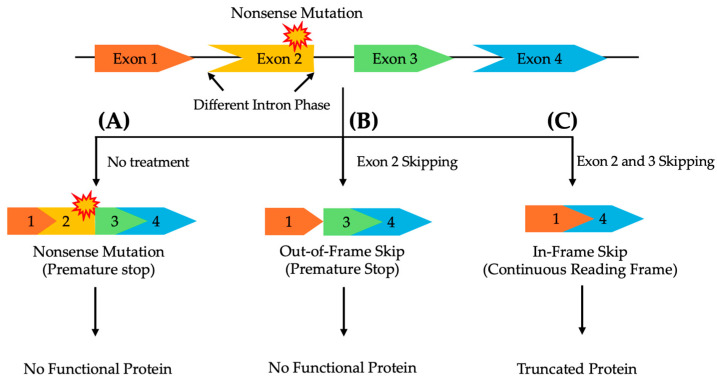
A schematic representation of multi-exon skipping to exclude pathogenic variants in asymmetrical exons, flanked by differing intron phases, while preserving the reading frame to enable the production of a functional protein. (**A**) Without treatment, a nonsense mutation in exon 2 prevents the synthesis of a functional protein. (**B**) Skipping only exon 2 disrupts the reading frame, leading to a non-functional protein. (**C**) Skipping both exons 2 and 3 restores the reading frame, resulting in a truncated, yet functional, protein.

**Figure 2 cells-14-00177-f002:**
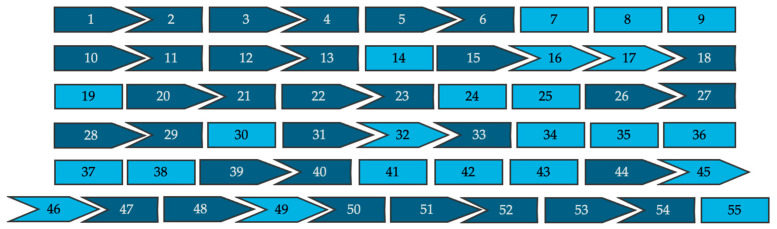
A visual representation of all 55 exons of the full-length dysferlin transcript and their respective intron phases (not to scale). Symmetrical exons are shown in light blue and asymmetrical exons are shown in dark blue.

**Figure 3 cells-14-00177-f003:**
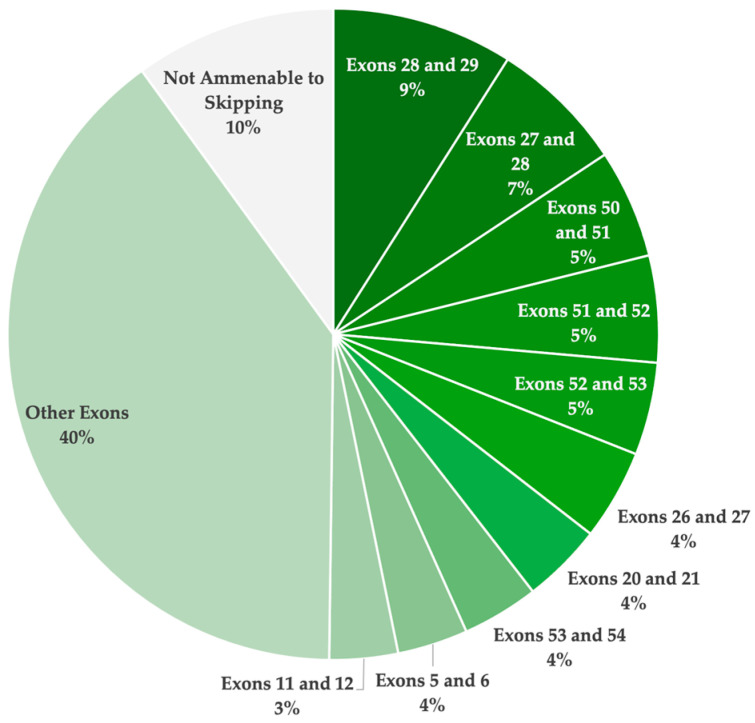
A pie chart illustrating the overall theoretical applicability of single- and double-exon skipping to exclude pathogenic mutations and/or restore the reading frame, resulting in a truncated yet potentially functional dysferlin protein. This analysis is based on the pathogenic variants reported in the UMD-DYSF database. The ten most highly applicable targets are highlighted.

**Table 1 cells-14-00177-t001:** The frequency of disease-causing mutations in the *DYSF* gene as reported in the UMD-DYSF database.

Mutation Type	Overall Frequency
**Exonic Point Mutations**	**54%**
Nonsense Mutations	26%
Missense Mutations	28%
**Exonic Insertions/Deletions**	**29%**
**Intronic Variants**	**17%**

**Table 2 cells-14-00177-t002:** An overview of the applicability of all single- and double-exon skipping approaches for all pathogenic variants reported in patients with dysferlinopathies in the UMD-DYSF database. Each exon skipping strategy was evaluated and ranked based on its potential applicability to the overall patient population analyzed. Exons previously validated in vitro as effective targets for ASO-mediated exon skipping are highlighted in bold. Exons previously identified to contain critical regions essential to dysferlin protein function, which are predicted to yield a non-functional product if skipped, are marked with an asterisk (*).

Ranking	Target Exon(s)	Overall Applicability
1	**28 and 29**	9.0%
2	27 and 28	6.7%
3	50 and 51	5.4%
4	51 and 52	5.3%
5	52 and 53	4.6%
6	26 and 27	4.5%
7	20 and 21 *	4.0%
8	53 and 54	3.8%
9	5 and 6	3.5%
10	11 and 12	3.4%
11	11	2.9%
12	39 and 40	2.8%
13	24	2.8%
14	12 and 13	2.7%
15	37	2.6%
16	25 *	2.6%
17	19 *	2.6%
18	49	2.4%
19	8	2.4%
20	22	2.2%
21	4 and 5	2.1%
22	3 and 4	2.0%
23	43	1.8%
24	**32**	1.8%
25	34	1.7%
26	47 and 48 *	1.6%
27	45 *	1.6%
28	10 and 11	1.5%
29	41	1.5%
30	38	1.5%
31	30	1.5%
32	7	1.4%
33	21 and 22 *	1.2%
34	2 and 3	1.2%
35	13	1.2%
38	52	1.1%
37	50	1.1%
36	12	1.1%
39	46 *	1.0%
40	16	1.0%
41	5	1.0%
42	22 and 23	0.8%
43	42	0.8%
44	27	0.8%
45	36	0.6%
46	14	0.6%
47	3	0.5%
48	48 *	0.4%
49	51	0.3%
50	10	0.3%
58	31 and 32	0.2%
56	19 and 20 *	0.2%
55	13 and 14	0.2%
51	35	0.2%
57	29	0.2%
52	20 *	0.2%
54	6	0.2%
53	4	0.2%
59	53	0.1%
60	47 *	0.1%
61	39	0.1%

**Table 3 cells-14-00177-t003:** A summary of the applicability of single- and double-exon skipping for all 962 pathogenic variants reported in patients in the UMD-DYSF database.

	Applicable Variants	Total Applicability
Single-Exon Skipping	430	44.6%
Double-Exon Skipping	436	45.3%
Single- and/or Double-Exon Skipping	866	90.0%

## Data Availability

The dataset analyzed in this study is available in the UMD-DMD database at http://www.umd.be/DYSF/. The original contributions presented in this study are included in the article. Further inquiries can be directed to the corresponding author.

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
