# Peer review of "Analysis of Exon Skipping Applicability for Dysferlinopathies"

_cells, 2025, doi:10.3390/cells14030177_

Round 1
Reviewer 1 Report
Comments and Suggestions for Authors
This is a well-written review about an important area of translational research, but attention to a few key points would greatly improve it. In general, this review should be set in a more comprehensive, realistic, and up-to-date context. Specific suggestions are as follows:
1. Lines 115-116: The authors mention that AAV-mediated gene therapy is underway as a potential treatment strategy for dysferlinopathies. However, the two studies cited are from 2010 & 2015, and there are other more recent studies worth citing (e.g., Potter et al, HGT, 2018). It would be helpful if the authors provided some insight into why this field is lagging so far behind DMD, in terms of understanding disease mechanisms and implementing therapeutic strategies. There appear to be only two published reports (one of which has been retracted) testing functionality of truncated forms of dysferlin (Azakir et al, J Biol Chem, 2012; Llanga et al, Mol Ther, 2017) - why isn't this strategy being pursued? Some brief, up-to-date information about the difficulties in this field should be included to provide the proper context for a discussion of ASO strategies. If gene therapy options have been largely abandoned, that is even more reason to pursue the translation of ASOs. However, if it is true that only full-length dysferlin can correct pathology, this needs to be stated as a caveat of the ASO approach.
2. Line 125: " ... alter the transcript’s composition" is confusing; not sure what is meant by this.
3. Lines 175-179: "Notably, several ASOs have been approved for the treatment of DMD, including Eteplirsen, Viltolarsen, Golodirsen, and Casimersen [80]. By promoting the exclusion of specific exons, these therapies have demonstrated effectiveness in restoring partially functional dystrophin protein levels in patients who otherwise lack functional dystrophin [79,81]." These old references from 2017 and 2018 are very misleading, as these DMD drugs are generally considered to be a failure (e.g., Hong et al, JAMA Network Open, 2024). This section should be updated with a more accurate take on the clinical (lack of) effectiveness of approved exon-skipping treatments for DMD and, by extension, a more realistic outlook on the likely efficacy of similar treatments for the dysferlinopathies.
4. Table 1: It would be very helpful and informative if this table included information on which exons have been successfully (or unsuccessfully) targeted in preclinical studies to date. If any skipping strategies are predicted to yield a product that has been shown to be partly functional (as with some naturally occuring variants), that would be useful to mention in the table.
Author Response
Response to Reviewer One:
This is a well-written review about an important area of translational research, but attention to a few key points would greatly improve it. In general, this review should be set in a more comprehensive, realistic, and up-to-date context. Specific suggestions are as follows:
Comment 1: Lines 115-116: The authors mention that AAV-mediated gene therapy is underway as a potential treatment strategy for dysferlinopathies. However, the two studies cited are from 2010 & 2015, and there are other more recent studies worth citing (e.g., Potter et al, HGT, 2018). It would be helpful if the authors provided some insight into why this field is lagging so far behind DMD, in terms of understanding disease mechanisms and implementing therapeutic strategies. There appear to be only two published reports (one of which has been retracted) testing functionality of truncated forms of dysferlin (Azakir et al, J Biol Chem, 2012; Llanga et al, Mol Ther, 2017) - why isn't this strategy being pursued? Some brief, up-to-date information about the difficulties in this field should be included to provide the proper context for a discussion of ASO strategies. If gene therapy options have been largely abandoned, that is even more reason to pursue the translation of ASOs. However, if it is true that only full-length dysferlin can correct pathology, this needs to be stated as a caveat of the ASO approach.
Response: We greatly appreciate your comment acknowledging the absence of an up-to-date discussion of the limitations associated with AAV-mediated gene therapy for dysferlinopathies, as this insight underscores the importance of exploring alternative therapeutic approaches. This section has been updated and now includes a brief discussion on lines 114-128 that reads: “However, numerous studies have focused on evaluating a range of different molecular and genetic therapies aiming to improve the patient’s phenotype and prevent progression [51]. These approaches include strategies like gene therapy, delivering a modified dysferlin gene using AAV vectors [52–54], and gene editing, using CRISPR-Cas9 systems to correct specific DYSF variants [55,56]. However, the AAV vectors used to delivery these therapies have significant limitations and safety concerns. Natural exposure to AAVs has resulted in 30-60% of the population developing antibodies against these vectors, which can compromise their effectiveness and raise the risk of adverse immune reactions [57–59]. Additional safety risks include the development of innate and adaptive immune responses and the potential genomic consequences of viral genome insertion exist [60,61]. Moreover, despite their therapeutic potential, several AAV-related patient fatalities, often associated with high doses, have raised concerns about the clinical safety of these vectors [62,63]. Alternatively, ASO-mediated exon-skipping, which uses non-viral approaches to exclude pathogenic variants from the final transcript, shows significant promise and will be discussed in further detail.”
Comment 2: Line 125: " ... alter the transcript’s composition" is confusing; not sure what is meant by this.
Response: We appreciate the valuable comment. The sentence from line 125 has been updated from “alter the transcript’s composition” to “alter the transcript’s splicing” for improved clarity.
Comment 3: Lines 175-179: "Notably, several ASOs have been approved for the treatment of DMD, including Eteplirsen, Viltolarsen, Golodirsen, and Casimersen [80]. By promoting the exclusion of specific exons, these therapies have demonstrated effectiveness in restoring partially functional dystrophin protein levels in patients who otherwise lack functional dystrophin [79,81]." These old references from 2017 and 2018 are very misleading, as these DMD drugs are generally considered to be a failure (e.g., Hong et al, JAMA Network Open, 2024). This section should be updated with a more accurate take on the clinical (lack of) effectiveness of approved exon-skipping treatments for DMD and, by extension, a more realistic outlook on the likely efficacy of similar treatments for the dysferlinopathies.
Response: We are grateful for your thoughtful input. The section on Lines 194-200 have been updated to include a more accurate discussion on the effectiveness of approved exon-skipping treatments for DMD that reads: “Despite their potential, the clinical effectiveness of these therapies for DMD remains controversial at this time. Treated patients typically show only modest increases in dystrophin expression, and the clinical significance of the minimal functional benefits observed is difficult to determine due to small patient cohorts and reliance on historical controls [79,89,90]. Enhancing ASO delivery and cellular uptake will be essential to improving their therapeutic effect [67].”
Comment 4: Table 1: It would be very helpful and informative if this table included information on which exons have been successfully (or unsuccessfully) targeted in preclinical studies to date. If any skipping strategies are predicted to yield a product that has been shown to be partly functional (as with some naturally occurring variants), that would be useful to mention in the table.
Response: Thank you for suggesting the inclusion of information on exons predicted to be suitable or unsuitable for exon-skipping in Table 2 (originally Table 1). The table has been updated to have exon-skipping approaches previously validated shown in bold and those predicted to produce a non-functional product indicated by an asterisk. The figure caption has been updated to reflect these changes and reads: “Exons previously validated in vitro as effective targets for ASO-mediated exon-skipping are highlighted in bold. Exons previously identified to contain critical regions essential to DYSF protein function, which are predicted to yield a nonfunctional product if skipped, are marked with an asterisk (*).”
Reviewer 2 Report
Comments and Suggestions for Authors
The manuscript “Analysis of Exon-Skipping Applicability for Dysferlinopathies” reports on potential use of ASO-mediated exon skipping as a therapeutic strategy for dysferlinopathies. The review is concise but very clear and well written. Some points need to be better discussed to help non-expert readers to follow the data.
1. Table 1: it is not clear which criteria have been applied for exons skipping strategy ranking. Can the authors discuss in more detail this point?
2. Figure 3: can the authors discuss in more detail how did they evaluate clinical applicability of the exon skipping strategy.
3. Adding an additional figure depicting the outcome of exon skipping on protein structure (at least for some of the most significant application) may improve the prediction of the possible impact of this strategy on protein function. Are there key functional domains in the proteins that absolutely need to be preserved?
4. Can the authors comment on the advantage of using an ASO mediated exon skipping strategy compared to CRISPR-CAS9?
Author Response
Response to Reviewer Two:
The manuscript “Analysis of Exon-Skipping Applicability for Dysferlinopathies” reports on potential use of ASO-mediated exon skipping as a therapeutic strategy for dysferlinopathies. The review is concise but very clear and well written. Some points need to be better discussed to help non-expert readers to follow the data.
Comment 1: Table 1: it is not clear which criteria have been applied for exons skipping strategy ranking. Can the authors discuss in more detail this point?
Response: We appreciate the valuable comment to include a description of how the exon-skipping strategies were ranked. A description of the ranking of these strategies has been included in Table 2’s (originally Table 1) caption, that reads: “Each exon-skipping strategy was evaluated and ranked based on its potential applicability to the overall patient population analyzed”
Comment 2: Figure 3: can the authors discuss in more detail how did they evaluate clinical applicability of the exon skipping strategy.
Response: Thank you for your suggestion. This figure provides an overview of the theoretical applicability of all exon-skipping strategies identified. To clarify that this figure is based on theoretical applicability, instead of clinical applicability, the figure caption has been updated and reads: “Pie chart illustrating the overall theoretical applicability of single and double exon-skipping to exclude pathogenic mutations and/or restore the reading frame, resulting in a truncated yet potentially functional dysferlin protein. This analysis is based on the pathogenic variants reported in the UMD-DYSF database. The ten most highly applicable targets are highlighted.”
Comment 3: Adding an additional figure depicting the outcome of exon skipping on protein structure (at least for some of the most significant application) may improve the prediction of the possible impact of this strategy on protein function. Are there key functional domains in the proteins that absolutely need to be preserved?
Response: We appreciate your thoughtful comment. For improved clarity, the discussion of the critical regions of the dysferlin protein has been relocated from section 4.3. to section 3.1. on lines 238-247 that reads: “A critical challenge associated with the effectiveness of these approaches is the possibility that certain exons contain essential regions, and their exclusion may result in a non-functional or improperly folded protein, unable to restore function [87,93]. For instance, while skipping exon 25 maintains the reading frame, it still produces an unstable product that fails to yield a functional protein [94]. Additionally, plasma-membrane-resealing assays have identified exons 19 to 21, 20 to 21, and 46 to 48 as crucial regions for dysferlin-associated membrane repair [95]. Thus, excluding these regions is unlikely to improve symptoms in patients. When feasible, analyzing previously reported in-frame large deletions in patient populations related to the target regions for exclusion may help predict the functionality of the truncated protein generated through exon-skipping.” In addition, Table 2 has been updated to highlight exon-skipping strategies in these critical regions. Table 2’s caption has been updated to the following: “Exons previously identified to contain critical regions essential to dysferlin protein function, which are predicted to yield a nonfunctional product if skipped, are marked with an asterisk (*).”
Comment 4: Can the authors comment on the advantage of using an ASO mediated exon skipping strategy compared to CRISPR-CAS9?
Response: Thank you for the valuable suggestion to discuss the advantages of ASO-mediated exon-skipping compared to CRISPR-CAS9. The following description has been included on lines 118-128 that reads: “However, the AAV vectors used to delivery these therapies have significant limitations and safety concerns. Natural exposure to AAVs has resulted in 30-60% of the population developing antibodies against these vectors, which can compromise their effectiveness and raise the risk of adverse immune reactions [57–59]. Additional safety risks include the development of innate and adaptive immune responses and the potential genomic consequences of viral genome insertion exist [60,61]. Moreover, despite their therapeutic potential, several AAV-related patient fatalities, often associated with high doses, have raised concerns about the clinical safety of these vectors [62,63]. Alternatively, ASO-mediated exon-skipping, which uses non-viral approaches to exclude pathogenic variants from the final transcript, shows significant promise and will be discussed in further detail.”
Reviewer 3 Report
Comments and Suggestions for Authors
In this manuscript the authors provide an in silico analysis of the potential application of exon-skipping therapy to Dysferlinopathies. The manuscript is well-structured and reports novel information that has relevance for future studies.
However it will be important that the authors further comment on:
- Limitations of exon skipping therapies – in this point it would be important to extend the section “3.1. ASO-Mediated Exon-Skipping for the Treatment of Dysferlinopathies” further commenting on the functionality of truncated versions of dysferlin
- Details on the mutation types found in DYSF gene – a table summarizing the percentage of silent, missense and non-sense mutations among the variants found is very important, since the applicability of exon skipping to non-sense mutations will be the most important
- The multiple exons that are altered in DYSF – would this still make it feasible to use exon-skipping as the right approach?
Minor comments:
Line 12: DYSF gene should be italicized
Author Response
Response to Reviewer Three:
In this manuscript the authors provide an in silico analysis of the potential application of exon-skipping therapy to Dysferlinopathies. The manuscript is well-structured and reports novel information that has relevance for future studies.
Comment 1: Limitations of exon skipping therapies – in this point it would be important to extend the section “3.1. ASO-Mediated Exon-Skipping for the Treatment of Dysferlinopathies” further commenting on the functionality of truncated versions of dysferlin
Response: Thank you for your insightful comment regarding the inclusion of a discussion on the functionality of truncated dysferlin in Section 3.1. This topic was originally addressed in Section 4.3. but has now been relocated to Section 3.1. for improved flow. The discussion can now be found on lines 238-247 that reads: “A critical challenge associated with the effectiveness of these approaches is the possibility that certain exons contain essential regions, and their exclusion may result in a non-functional or improperly folded protein, unable to restore function [87,93]. For instance, while skipping exon 25 maintains the reading frame, it still produces an unstable product that fails to yield a functional protein [94]. Additionally, plasma-membrane-resealing assays have identified exons 19 to 21, 20 to 21, and 46 to 48 as crucial regions for dysferlin-associated membrane repair [95]. Thus, excluding these regions is unlikely to improve symptoms in patients. When feasible, analyzing previously reported in-frame large deletions in patient populations related to the target regions for exclusion may help predict the functionality of the truncated protein generated through exon-skipping.”
Comment 2: Details on the mutation types found in DYSF gene – a table summarizing the percentage of silent, missense and non-sense mutations among the variants found is very important, since the applicability of exon skipping to non-sense mutations will be the most important
Response: We appreciate your valuable suggestion. On line 249, a description of the percentage of point mutations that are nonsense variants and missense variants was included that reads: “Among the point mutations, 49% are nonsense variants, while 51% are missense variants.” In addition, Table 1 has been created to outline the frequency of different mutation types associated with dysferlinopathies.
Comment 3: The multiple exons that are altered in DYSF – would this still make it feasible to use exon-skipping as the right approach?
Response: Thank you for this thoughtful suggestion. A discussion of the feasibility of exon-skipping approaches for multi-exon deletions has been included on lines 217-220 that reads: “Furthermore, in cases involving large multi-exon deletions in the DYSF gene [98], a strategy similar to current DMD therapies could be applied, where promoting the skipping of adjacent exons can restore the DYSF reading frame [99], potentially enabling the production of a partially functional dysferlin protein.”
Comment 4: Line 12: DYSF gene should be italicized
Response: We appreciate your comment. DYSF on line 12 has been italicized.